# Trends in Mortality Due to Malignant Neoplasms of Female Genital Organs in Poland in the Period 2000–2021—A Population-Based Study

**DOI:** 10.3390/cancers16051038

**Published:** 2024-03-03

**Authors:** Małgorzata Pikala, Monika Burzyńska

**Affiliations:** Department of Epidemiology and Biostatistics, The Chair of Social and Preventive Medicine of the Medical University of Lodz, 90-752 Lodz, Poland; monika.burzynska@umed.lodz.pl

**Keywords:** mortality, cancer, trends, female genital organs, epidemiology, Poland

## Abstract

**Simple Summary:**

Malignant neoplasms of female genital organs cause a large number of deaths worldwide. The death rates from these diseases are higher in Poland than in many European Union countries. Educational programmes, screening tests and HPV vaccination are being introduced in Poland to counteract these differences. Monitoring the effectiveness of the planned actions should be accompanied by the implementation of studies on changes in mortality trends. In our study, we analysed all deaths of female residents of Poland between 2000 and 2021 included in the Statistics Poland database. We determined trends in changes in mortality due to malignant neoplasms of female genital organs in total and separately for individual cancers from this group. We also analysed educational and urbanisation differences in mortality due to this group of tumours. The obtained results should be useful in designing adequate preventive actions.

**Abstract:**

The aim of this study is to assess mortality trends due to malignant neoplasms of female genital organs (MNFGOs) in Poland between 2000 and 2021. For the purpose of the study, the authors used data on all deaths of Polish female inhabitants due to MNFGO between 2000 and 2021, obtained from the Statistics Poland database. The standardised death rates (SDR), potential years of life lost (PYLL), annual percentage change (APC) and average annual percentage change (AAPC) were calculated. Between the years 2000 and 2021, 138,000 women died due to MNFGOs in Poland. Of this number, 54,975 (39.8%) deaths were caused by ovarian cancer, 37,487 (27.2%) by cervix uteri cancer, and 26,231 (19.0%) by corpus uteri cancer. A decrease in mortality due to cervix uteri cancer (APC = −2.4%, *p* < 0.05) was the most favourable change that occurred in the period 2000–2021, while the least favourable change was an increase in mortality due to corpus uteri cancer for the period 2005–2019 (APC = 5.0%, *p* < 0.05). SDRs due to ovarian cancer showed a decreasing trend between 2007 and 2021 (APC = −0.5%, *p* < 0.05). The standardised PYLL index due to cervical cancer was 167.7 per 100,000 women in 2000 and decreased to 75.0 in 2021 (AAPC = −3.7, *p* < 0.05). The number of lost years of life due to ovarian cancer decreased from 143.8 in 2000 to 109.5 in 2021 (AAPC = −1.3, *p* < 0.05). High values of death rates due to MNFGO in Poland, compared to other European countries, show that there is a need to promote preventive programmes and continue to monitor changes in mortality.

## 1. Introduction

In 2020, 4.4 million women died from malignant neoplasms worldwide. Malignant neoplasms of female genital organs (MNFGO) were responsible for approximately 672,000 deaths. This represented 15.3% of all cancer deaths among women. The most common causes of death in the MNFGO group were cervix uteri cancer (7.8%), ovarian cancer (4.7%) and corpus uteri cancer (2.2%) [1,2].

MNFGOs are not only an important health problem but also a social problem because mortality due to these causes begins to increase from the age of 35 and often concerns women, professionally active and raising children. These diseases are also important from an economic point of view as both direct costs, including diagnosis, treatment, and prevention, as well as indirect costs, associated with patients’ disability, premature death and lost years of life, are very high [3,4,5,6]. The latest results from the Global Burden of Diseases, Injuries, and Risk Factors Study, covering the period 2000–2019, reveal that MNFGOs are leading cancers contributing to disability-adjusted years of life years. This disease group includes cervical cancer, ovarian cancer and corpus uteri cancer, which, respectively, constitute the fourth, sixth and thirteenth cause of mortality [7].

In the year 2020, 6811 women died from MNFGO in Poland. This number accounted for 15.0% of all cancer deaths among Polish women. The three most common causes of death in the MNFGO group were ovarian cancer (5.9%), corpus uteri cancer (4.0%) and cervix uteri cancer (3.3%).

The standardised death rates (SDRs) due to MNFGOs in Poland are some of the highest among all European Union (EU) countries. In 2020, the SDR value due to ovarian cancer, calculated per 100,000 women, was 9.6 for the 27 EU countries and 12.9 for Poland. Higher SDRs values were observed in only four countries: Malta (13.3), Ireland (13.9), Lithuania (14.5) and Latvia (17.1) (Figure 1a) [8]. In 2020, the SDR value due to malignant neoplasm of other parts of uterus was 6.6 in the EU and 9.4 in Poland. Worse results were noted only in Latvia, where the SDR was 9.6 (Figure 1b).

In 2020, the SDR value due to cervix uteri cancer was 3.7 in the EU and 7.2 in Poland. For this cancer, higher SDR values were observed in five other EU countries: Estonia (7.9), Bulgaria (9.0), Latvia (9.2), Lithuania (11.0) and Romania (12.7) (Figure 1c).

Over the past two decades, the survival of Polish patients with MNFGOs has been steadily improving. Age-standardised 5-year net survival (NS) has increased for almost all cancer types in this group. The largest statistically significant increase was observed for ovarian cancer, i.e., from 35.7% to 41.3% (by 5.6 percentage points), followed by cervical cancer, where NS increased from 52.7% to 56.9% (by 4.2 percentage points). For corpus uteri cancer, NS increased from 72.2% to 76.0% (by 3.8 percentage points) [9]. However, a comparison analysis of MNFGO survival rates observed in Poland and those noted in other European countries, such as Norway and Finland, shows huge significant differences. The most significant difference was observed for cervix uteri cancer (−16.2 percentage points compared to Finland and −25.1 percentage points compared to Norway) [10,11].

The unfavourable differences in death rates and survival rates in Poland compared to other EU countries indicate that certain measures have to be implemented to improve the situation in this area.

Since the mid-2000s, several attempts have been made to improve health outcomes for cancer patients in Poland, including the adoption of the National Programme for Cancer Diseases Control for 2006–2015 [12], renewed for 2016–2024. However, these programs focused on strengthening preventative measures, including increasing participation in cancer screening programs and improving access to diagnostics and treatment, rather than on improving the organisation of care. The solution aimed at improving the healthcare system in this aspect was the introduction (in 2015) of a dedicated ‘fast pathway’ for patients with suspected cancer to enable them faster access to comprehensive diagnostics and treatment [13,14] and the piloting (from 2019) of the National Oncology Network, which is one of the part of the National Cancer Strategy 2020–2030 [15], aiming at increasing the number of people with 5-year survival after completing cancer therapy as well as reducing the incidence of neoplasms. Increased detectability of cancers in early stages and reduced mortality should be achieved. For cervical cancer, a target mortality rate of 4.9 in 2025 has been specified. It is also assumed that the percentage of individuals participating in cervical cancer screening will increase to 80% by the end of 2027. The effect will be achieved by including primary care physicians and occupational medicine specialists in secondary prevention efforts, increasing the quality control of cytological tests, and tightening the criteria for implementing the “Cervical Cancer Screening Program”. As part of the oncology strategy, care is also provided for families at high, genetically determined risk of developing selected malignant tumours, including ovarian and corpus uteri cancers. In the first stage, the identification of individuals with a high, hereditary predisposition to certain cancers takes place. A detailed family history plays a crucial role here. If there are medical indications, genetic testing is performed. In the second stage, individuals with a high, hereditary predisposition to cancer undergo specialised supervision, involving systematic diagnostic testing and medical consultations. The National Oncology Strategy envisages the continuation of tasks from previous programmes as part of efforts to reverse unfavourable epidemiological trends and reduce indirect costs resulting from the burden of cancer diseases, including MNFGO. According to the European Cancer Plan, the strategy aims to increase investment in the medical workforce, lifestyle education, patient care, research, and innovation, as well as the oncological care system through the normalisation of diagnostic and therapeutic pathways. The strategy also includes further data collection from cancer registries to monitor the quality of care [16].

Evaluating the effectiveness of the planned actions should be accompanied by the implementation of studies both on changes in mortality trends and on the impact of sociodemographic variables on mortality from malignant neoplasms in Poland.

The present study aims to assess mortality trends due to MNFGOs in Poland between the years 2000 and 2021, considering differences related to education and place of residence (urban areas vs. rural areas). The results of these analyses should eventually contribute to implementation of actions necessary to reduce health inequalities between Poland and other European countries.

## 2. Materials and Methods

The study material was a database including 138,000 death certificates of all Polish women who died due to MNFGOs in the period 2000–2021 (according to the International Statistical Classification of Diseases and Health Related Problems—Tenth Revision—ICD-10, coded as C51–C58). In the statistical analysis, the following causes of death due to MNFGO were identified: vulvar cancer (C51), vaginal cancer (C52), cervical cancer (C53), uterine corpus cancer (C54), unspecified uterine cancer (C55), ovarian cancer (C56) and cancer of other female genital organs (C57). Due to a low number of placental cancer cases (C58), this cancer type was excluded from any statistical analysis. The data were provided by Statistics Poland.

In Poland, as in most countries in the world, the statistical system uses the so-called primary cause of death, which is entered into the death certificate. The primary cause is referred to as a disease contributing to a pathological process, which in turn led to death.

The authors calculated standardised death rates (SDRs), and the standardisation was carried out using the European Standard Population, updated in 2012 [17].

Potential years of life lost (PYLL) due to the three leading causes of death in the MNFGO group were also calculated for the years 2000–2021. This indicator is a summary measure of premature mortality, providing an explicit way of weighting deaths occurring at younger ages, which may be preventable. The calculation of potential years of life lost involves summing up deaths occurring at each age and multiplying this with the number of remaining years to live up to a selected age limit (age 75 was adopted in accordance with OECD Health Statistics) [18].

The data on deaths by educational level were obtained from records of death certificates via the database of the Central Statistical Office in Poland, whereas data for population denominators for the same educational categories were retrieved from the census. Educational categories refer to the attained educational level. The Polish educational scheme has been reclassified into three categories corresponding to the International Standard Classification of Education (ISCED 2011): lower secondary education or less (categories 0–2), upper secondary education (categories 3–4) and higher education (categories 5–6) [19].

The place of residence of deceased is coded in death certificates according to the system used by the Central Statistical Office [20].

To assess differences in mortality related to education and place of residence (urban vs rural), inequality indices were calculated as the ratio of death rates in respective groups [21].

The Joinpoint Regression Program, designed by the U.S. National Cancer Institute for the Surveillance, Epidemiology and End Results Program, was used in order to analyse mortality trends [22]. The authors also calculated the annual percentage change (APC) and average annual percent change (AAPC) along with 95% confidence intervals (CI).

The method of calculation of the above rates has been described in previous publications [6,23].

The calculations and graphs were performed using Statistica, version 13 (TIBCO Software Inc., Santa Clara, CA, USA).

## 3. Results

Between 2000 and 2021, 138,000 women died from malignant neoplasms of female genital organs (MNFGO) in Poland. The three most common causes of death in the MNFGO group were ovarian cancer, responsible for 54,975 deaths between 2000 and 2021; cervix uteri cancer, responsible for 37,487 deaths; and corpus uteri cancer, responsible for 26,231 deaths (Appendix A).

The standardised death rate (SDR) due to MNFGO in 2000 was 36.7 per 100,000 women. In 2021, the SDR decreased to 30.1 per 100,000. The average annual percentage change (AAPC) between 2000 and 2021 was −0.8 (*p* < 0.05) (Table 1 and Appendix A).

The most favourable changes among the three leading causes of death in the MNFGO group were for cervix uteri cancer. The SDR almost halved over the analysis period, from 12.0 in 2000 to 6.4 in 2021 (APC = −2.4; *p* < 0.05) (Appendix A, Table 1, Figure 2).

For ovarian cancer, two opposing trends were observed. Between 2000 and 2007, the SDR increased from 12.7 to 13.9 (APC = 1.1%; *p* < 0.05); between 2007 and 2021, there was a decrease in SDR from 13.9 to 12.5 (APC = −0.5%; *p* < 0.05).

The SDR due to corpus uteri cancer increased from 5.3 in 2000 to 7.6 in 2021 (AAPC = 1.9, *p* < 0.05). A statistically significant change occurred between the years 2005 and 2019. During this period, the standardised death rate (SDR) increased at a rate of 5% annually. It is worth noting that in 2016 there was an intersection of mortality trends for cervix uteri cancer and corpus uteri cancer. Since that year, higher SDR values have occurred for the latter cancer.

Among the less common causes of death in the MNFGO group, there was a decrease in SDR for uterus unspecified cancer (APC = −8.0; *p* < 0.05), a decrease in SDR for vagina cancer (APC = −2.9; *p* < 0.05), and an increase in SDR for vulva cancer (APC = 1.2, *p* < 0.05).

Premature mortality (below the age of 75) is a direct cause of lost years of life. The standardised PYLL index due to cervical cancer was 167.7 per 100,000 women in 2000 and decreased to 75.0 in 2021 (AAPC = −3.7, *p* < 0.05) (Table 2). The number of lost years of life due to ovarian cancer also decreased from 143.8 in 2000 to 109.5 in 2021 (AAPC = −1.3, *p* < 0.05). For corpus uteri cancer, periodic increases and decreases of the PYLL index were observed. Over the entire analysed period, a statistically significant increase was noted in the average annual percentage change, amounting to 1.3%.

Educational and urbanisation differences in death rates for MNFGO overall and for the three most common causes of death in this group in 2021 were analysed. Among the 6415 women who died in Poland in 2021 due to MNFGO, 728 had higher education (11.3%), 3667 had secondary education (57.2%), and 1814 had primary education (28.3%). Information on education was missing in 206 death certificates (3.2%). A total of 4130 deceased women lived in urban areas (64.4%), while 2285 deceased women lived in rural areas (35.6%). As a result of this analysis, large differences were observed in the values of death rates between the three educational groups: university, secondary and elementary. The lowest values of these rates were for women with university education, while the highest values were for the group of women with elementary education (Figure 3a). The MNFGO death rate for women with primary education was 69.9 per 100,000 women in 2021, 1.6 times higher than for women with secondary education and 4.4 times higher than for women with tertiary education. The highest educational inequalities were for corpus uteri cancer mortality. The death rate in the group of women with primary education was 20.8 and was 1.9 times higher than in the group with secondary education and 6.1 times higher than among women with tertiary education. The death rate due to cervix uteri cancer in the group of women with primary education was 14.6 and was 1.5 times higher than in the group with secondary education and 5.3 times higher than in the group with tertiary education. The smallest educational differences among the three main causes of MNFGO were for ovarian cancer. The death rate in the group of women with primary education was 16.5 and was 1.2 times higher than in the group of women with secondary education and 2.8 times higher than in the group with primary education.

Slight differences in death rates due to MNFGO were observed among women living in urban and rural areas. Slightly higher death rates occurred among urban residents than among rural residents. The death rate due to MNFGO in 2021 was 39.8 in the group of women living in urban areas and 35.9 in the group of women living in rural areas (Figure 3b). There were also slight differences to the disadvantage of urban female residents in mortality from the three most common causes of MNFGO deaths. The death rate among urban residents was 1.3 times higher than among rural residents for cervical cancer, 1.1 times higher for ovarian cancer, and 1.03 times higher for corpus uteri cancer.

## 4. Discussion

Changes in mortality trends due to MNFGO between 2000 and 2021 in Poland were characterised with different directions depending on the type of cancer. The most positive trends were observed for cervical cancer. The decline in mortality rates from this cause is associated with a decrease in the incidence of cervical cancer. Between 2000 and 2020, the standardised morbidity rates (SMR) decreased from 21.3 to 9.4 per 100,000 (AAPC = −3.9%) (Appendix A). The reduction in mortality was also influenced by the observed increase in survival rates from cervical cancer between 2000 and 2019. The overall age-standardised 5-year net survival has increased from 52.7% to 56.9% [9].

Cervical cancer is a special type of cancer. Due to full knowledge of its aetiology, it can be diagnosed early within screening programmes and prevented through vaccination. For this reason, cervical cancer is considered as a marker of public health and hereby particularly closely monitored. Many previous studies have shown an association between a decrease in cervical cancer mortality and an increase in socioeconomic levels, diminishing fertility rates, improved genital hygiene and a lower incidence of sexually transmitted diseases. Improvements in the aforementioned risk factors have resulted in decreased mortality due to cervical cancer in most European countries [24,25]. Following the implementation of cervical cancer screening programmes, death rates in many European countries started decreasing rapidly [26]. In Poland, in order to improve epidemiological data, the National Population Screening Programme for cervical cancer was developed and implemented in 2006. The programme included 9.7 million women aged 25–59, who were screened in a three-year interval. In 2010, a decrease of 5.7% in cervical cancer incidence and 3.4% in mortality was observed [27]. Our own study showed a further decrease in mortality rates at an average annual rate of −2.4%. However, the SDR value of 7.2 per 100,000 women in 2020 was still significantly higher than in Malta (1.7), Finland (1.5), Italy (1.4) and Luxembourg (0.6) (Figure 1a) [8]. Further action is therefore needed to reduce mortality due to cervical cancer. A greater number of people undergoing screening tests should be one such action. The Nordic countries have played a leading role within the international community. In Scandinavia, opportunistic screening occurred as early as the 1950s, and by 1996, all Nordic countries had implemented nationwide organised cytology-based screening [28]. As a result of these efforts, the incidence rates of cervical cancer have declined by 50–85% in the constituent countries over 50 years. However, while in the Nordic countries, the rates for cytology screening tests are as high as 70–80% [29], in Poland, this rate has always been very low, i.e., up to 30%. After the pandemic period, the rate decreased to less than 12% [30]. These figures may be slightly underestimated as they are reported by the National Health Fund and do not take into account cytological examinations performed in private surgeries. However, regardless of this, the number of preventive examinations is still too low. Simultaneously, the favourable trends in cervical cancer mortality over the past two decades are mainly attributed to the performance of screening tests. Various attempts at treatment did not improve survival rates [31]. Currently, there is an improvement in treatment outcomes with the use of immunotherapy [32], but these treatments are not reimbursed in Poland. Patients may benefit from the complicated and time-consuming procedure for emergency access to drug technologies (Polish RDTL) implemented in 2017 [33].

A second necessary measure for the prevention of cervical cancer in Poland is the introduction of vaccination against human papillomavirus (HPV). The authors of epidemiological studies revealed HPV DNA in 99.7 % of cervical cancer specimens, which confirms a direct causal link between HPV infection and cervical cancer [34]. In most European countries, HPV vaccination is widely available as part of national vaccination strategies, fully or partially funded by states [35]. In Poland, until 2023, vaccines were available free of charge to adolescents/children only as part of some prevention programmes run by local authorities. According to the plan in the National Oncology Strategy [16], HPV vaccination was supposed to be included in the Polish vaccination calendar in 2021, but the outbreak of the COVID-19 pandemic slowed down the process, and the recommended free-of-charge vaccination of adolescents of both sexes, aged 12–13 years, is going to be implemented only in the year 2023. It is difficult to determine now how many children will be vaccinated. However, surveys conducted on Polish parents show that many of them have a negative attitude towards HPV vaccination, fearing side effects and an impact on early sexual initiation [36].

The highest SDR values due to MNFGOs in Poland are observed for ovarian cancer. Despite the decreasing trend since 2007, SDR values are still much higher than those noted in the best European countries. In 2020, the SDR values were 12.9 in Poland, 7.5 in Spain, 6.6 in Iceland and 6.0 in Portugal (Figure 1b) [8]. These differences may be related, among other factors, to the lower prevalence of genetic mutations in these countries compared to Poland. It is mainly referred to mutations in the MLH1, MSH2, MSH6, EPCAM, and PMS2 genes, which predispose individuals to hereditary non-polyposis colorectal cancer (HNPCC), also known as Lynch syndrome, which significantly increase the risk of ovarian cancer [37,38,39]. Mutations in the BRCA1/2 genes are also relevant. Patients with this mutation represent approximately 20–25% of all patients with this type of cancer [40]. In Poland, in 2006, as part of the National Programme for Cancer Diseases Control, an effective program for the care of families with a high risk of hereditary cancer was introduced. Thanks to prophylactic removal of the ovaries and fallopian tubes, this is currently the only effective way to reduce the risk of ovarian cancer. We also observe progress in the treatment of the disease. In 2013, chemotherapy with bevacizumab in the first-line setting began to be financed under the Drug Program of the Ministry of Health [41]. These factors can be partly linked to the improvement in the ovarian cancer mortality rate after 2007.

Statistics show that ovarian cancer is characterised by the worst prognosis of all gynaecological cancers [42,43,44]. Due to a lack of early symptoms and effective prophylaxis, most patients come to the doctor at an advanced, metastatic stage, when the disease is no longer curable [45]. In the coming years, there is a chance for a further decrease in mortality due to the reduction in the incidence rate of ovarian cancer from 19.0 in 2000 to 14.9 in 2020 (AAPC = −1.2%) (Appendix A) and due to an increase in the overall age-standardised 5-year NS from 35.7% to 41.3% [9]. Key elements in the fight against ovarian cancer include access to molecular diagnostics, awareness of the presence of BRCA1 or BRCA2 gene mutations in the family, the use of effective treatment methods at an early stage of the disease, and improvement in the quality of comprehensive treatment.

The least favourable trends presented in our study were observed for mortality due to corpus uteri cancer. Between 2005 and the COVID-19 pandemic, SDR increased at a rapid annual rate of 5%. The reasons for the increase in mortality in this case are complex and not fully understood. Nevertheless, it seems plausible that the cause may be incomplete treatment of patients and too frequent abandonment of radiotherapy. The earlier treatment regimen involved the removal of the uterus and ovaries, followed by radiotherapy. Twenty-five years ago, reports began to emerge suggesting that radiotherapy could be omitted in some patients. It appears that in Poland, radiotherapy was too hastily abandoned in supplementary treatment for patients who were not fully correctly diagnosed. This is the most likely reason why the percentage of women surviving uterine cancer has been declining for some time. In 2005, the percentage of patients who died from corpus uteri cancer was 18%, similar to global standards. In 2018, it was 28%; in 2019, it was 31%; and in 2020, it was 35%. This may indicate that the abandonment of supplementary treatment is probably too widespread [46]. This unfavourable trend may also result from the fact that, in recent years, the profile of patients has changed—more and more younger women are being diagnosed. Many of them have irregular menstruation, and the first symptom of the disease, such as unexplained bleeding, is not alarming to them, unlike in menopausal women. Therefore, patients are diagnosed with more advanced stages of the disease [47].

Increased mortality caused by this cancer was also reported in many other countries. A total of 417,367 new cases and 97,370 new deaths of corpus uteri cancer were reported globally in 2020 [48]. Previous studies have revealed that higher corpus uteri cancer mortality rates occur in high-income countries and are associated with common risk factors such as smoking, drinking alcohol, physical inactivity, obesity, hypertension, diabetes and lipid disturbances [48,49]. The observed increase in mortality is worrying because early diagnosis of corpus uteri cancer is associated with a good prognosis. However, also in the case of this cancer, some positive changes have been observed in recent years. The incidence rate has started to decrease since 2016 (from 32.0 in 2016 to 25.3 in 2020; APC = −4.5%) (Appendix A). Additionally, an increase in the overall age-standardised 5-year NS from 72.2% to 76.0% has been observed [9].

According to OECD data, Poland belongs to the group of developed countries where premature mortality of the population and the associated high number of life years lost are observed [18]. In 2021, due to deaths before the age of 75, women in Poland lost 939,421 potential years of life (4765 per 100,000) [50]. The predominant cause of premature years of life lost is cancer, accounting for about 30% of PYLL in 2021. Tumours from the MNFGO group accounted for 5.2% of PYLL, a similar percentage to breast cancer (5.3% PYLL) and lung cancer (5.1%). Among the MNFGO group, ovarian cancer (2.4%), cervical cancer (1.6%), and corpus uteri cancer (0.8%) were the leading causes of PYLL.

Comparing the AAPC for SDR and PYLL index for cervical cancer and ovarian cancer reveals a faster rate of decline in PYLL. This indicates that the improvement in mortality due to these cancers is due to both a reduction in the number of deaths and later age at death.

The authors of the study also analysed educational and urbanisation differences in mortality due to MNFGOs in Poland. Huge differences were observed for SDRs between educational groups. The lowest values were noted for women with university education, whereas the highest ones were noted for women with primary education. Educational inequalities in the health status have been repeatedly studied and are well documented [51,52], also in studies on Poland [53]. The present study confirms these observations made for MNFGOs.

Results of the authors’ study confirm the disappearance of urbanisation differences in the health status of the Polish population observed in other studies, which is associated with increased access to new technologies and similar lifestyles of inhabitants of urban and rural areas [54]. In recent years, educational disparities between urban and rural residents have decreased, especially concerning individuals with the lowest level of education (in 2011, the percentage of individuals with primary education in rural areas was higher by 11.8 percentage points, whereas in 2022 it was higher by 6.9 percentage points), which may also contribute to reducing differences in health status.

In the study conducted in Poland between 2007 and 2009, an analysis on the participation of urban and rural women in the National Population-Based Cervical Cancer Screening Program was carried out. It was demonstrated that women living in rural areas were more likely to respond to invitations and participate in the screening compared to women living in urban areas (39.3% vs. 16.8%) [55]. Decreased mortality due to MNFGOs during the COVID-19 pandemic is noteworthy. A similar decrease was also observed for other malignancies in Poland and other European countries [56,57,58]. This may result from the fact that of two diseases a patient suffered, i.e., cancer and concurrent COVID-19, only the latter one was considered to contribute to the death. Results of other studies have shown that, during the pandemic, less people underwent cancer screening programmes in Poland than in the pre-pandemic period [59]. Only in subsequent years will we be able to determine the rate of increased cancer mortality, which was caused by avoiding preventive screening and making late diagnoses [60].

Furthermore, due to admission of almost 700,000 adult women from Ukraine (as of 21 August 2023) [61] and a much higher prevalence of cervix uteri cancer among female residents of this country [62], we might observe increased MNFGO-related mortality in Poland in subsequent years.

Preventable or potentially curable cancer is one of the most frequent causes of premature death in Europe. This problem poses a major challenge to Polish public health. One of the goals of the National Oncology Strategy 2020–2030 is reducing the incidence and mortality rates by reducing risk factors, investing in education and primary prevention. It will be crucial to improve awareness among the population of all age groups regarding the impact of healthy behaviours on cancer, as well as to implement legal regulations supporting healthy nutrition and anti-smoking policies. It will also be essential to engage medical staff, mainly primary care physicians, in primary prevention and active promotion of the principles of the European Code Against Cancer, as well as to continue reimbursing HPV vaccinations. To reverse unfavourable epidemiological trends, actions in the field of tertiary prevention are also necessary. This involves halting disease progression and limiting complications. In this regard, the strategy envisages the implementation of a new organisational model for patient care (“Cancer Units”) and the development of standards and guidelines for diagnostic and therapeutic procedures. This will be possible through the establishment of specialised treatment centres, improvement of diagnostics, enabling the use of effective therapies, and guiding patients through the entire process of diagnosis, treatment, and rehabilitation [12].

### Limitations

There are several limitations of this study. The quality of analyses based on death statistics depends on the completeness and reliability of information contained in death certificates. Death records in Poland are fully complete. However, the quality of coding of death causes is not fully satisfactory, but at this moment, there are no better source of the data in Poland.

Certain doubts may arise regarding deaths due to uterus unspecified cancer (C55), because this classification could encompass both corpus uteri cancer (C54) and cervix uteri cancer (C53). However, the number of deaths due to uterus unspecified cancer is relatively small—in 2021, it was 131 (2.0% of all deaths from the MNFGO group). Considering the comprehensive character of the study, which includes all cases of MNFGO-related deaths in Poland and the associated large dataset analysed, any potential improvement in coding causes of death would have had a minimal impact on the study conclusions.

## 5. Conclusions

High values of death rates due to MNFGO in Poland, compared to other European countries, show that there is a need to implement and promote adequate preventive programmes and educational activities, aimed primarily at less educated women.

Increasing mortality trends due to corpus uteri cancer call for searching for causes and further monitoring of the situation.

The fairly rapid increase in mortality rates observed from 2019, yet a statistically insignificant decrease due to the short observation period, calls for further observation of changes and the impact of the COVID-19 pandemic on MNFGO-related mortality.

## Figures and Tables

**Figure 1 cancers-16-01038-f001:**
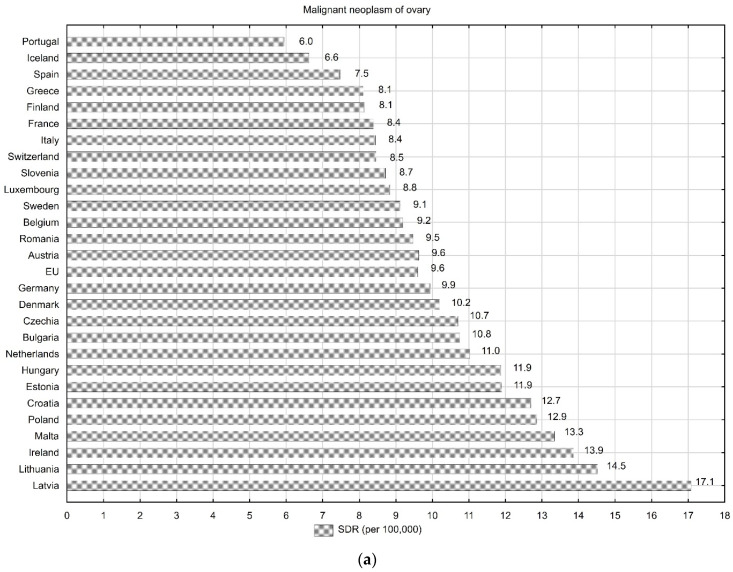
(**a**) Standardised death rates due to malignant neoplasm of ovary in EU countries in 2020. (**b**) Standardised death rates due to malignant neoplasm of other parts of uterus in EU countries in 2020. (**c**) Standardised death rates due to malignant neoplasm of cervix uteri in EU countries in 2020.

**Figure 2 cancers-16-01038-f002:**
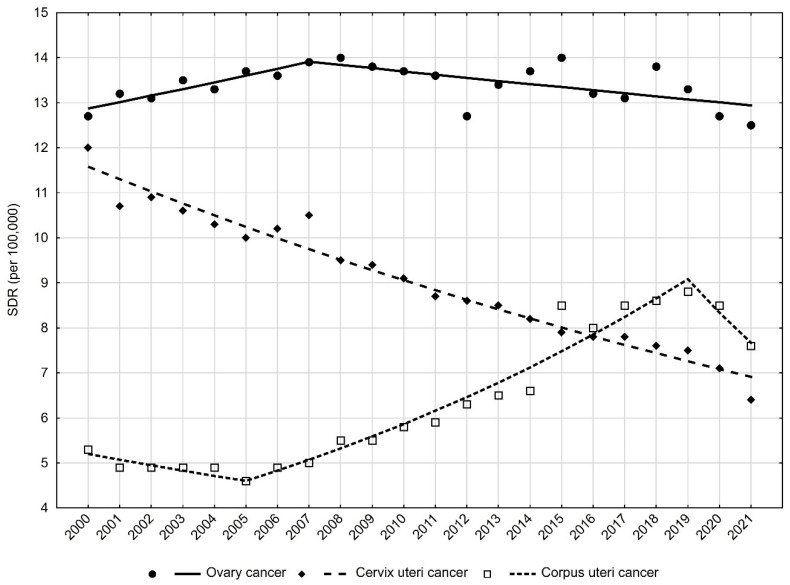
Mortality trends due to the most common malignant neoplasms of female genital organs in Poland in 2000–2021.

**Figure 3 cancers-16-01038-f003:**
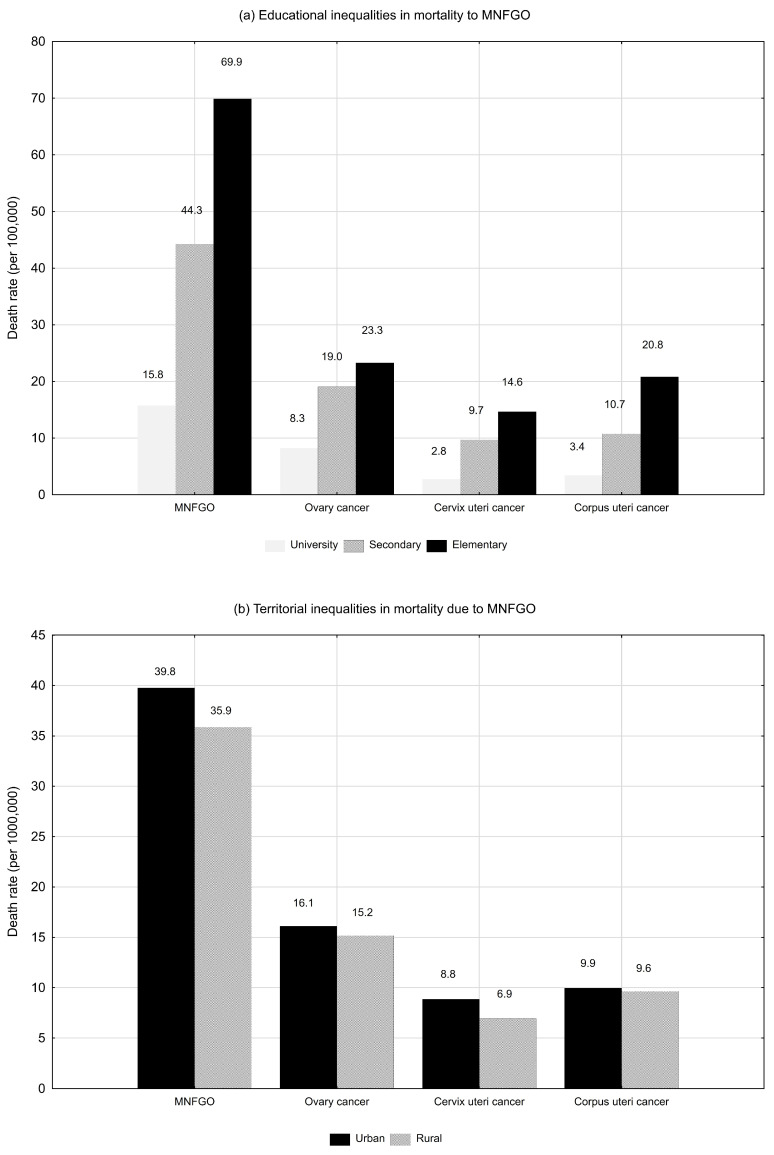
Educational and urbanisation differences in mortality due to malignant neoplasms of female genital organs in Poland in 2021.

**Table 1 cancers-16-01038-t001:** Time trends of standardised death rates (SDRs) due to malignant neoplasms of female genital organs in Poland in 2000–2021—joinpoint regression analysis.

Causes of Death	Number of Joinpoints	Years	APC (95% CI)	AAPC (95% CI)
Malignant neoplasms of female genital organs (C51–C58) including:	2	2000–2012 2012–2019 2019–2021	−0.8 * (−1.1; −0.4) 0.3 (−0.6; 1.2) −5.0 (−10.0; 0.2)	−0.8 *(−1.4; −0.3)
Vulva cancer (C51)	0	2000–2021	1.2 * (0,5; 2.0)	
Vagina cancer (C52)	0	2000–2021	−2.9 * (−4.4; −1.3)	
Cervix uteri cancer (C53)	0	2000–2021	−2.4 * (−2.6; −2.2)	
Corpus uteri cancer (C54)	2	2000–2005 2005–2019 2019–2021	−2.4 (−5.5; 0.8) 5.0 * (4.2; 5.8) −8.1 (−20.4; 6.2)	1.9 * (0.3; 3.4)
Uterus unspecified cancer (C55)	0	2000–2021	−8.0 * (−8.9; −7.2)	
Ovary cancer (C56)	1	2000–2007 2007–2021	1.1 * (0.0; 2.2) −0.5 * (−0.9; −0.1)	0.0 (−0.4; 0.4)
Other female genital organs cancer (C57)	2	2000–2014 2014–2017 2017–2021	−1.5 * (−2.5; −0.5) −14.5 (−31.8; 7.1) 5.1 (−2.2; 12.8)	−2.3 (−5.4; 0.9)

* *p* < 0.05.

**Table 2 cancers-16-01038-t002:** Standardised potential life years lost for women due to the three most common causes of death in the MNFGO group in the years 2000–2021 (per 100,000 population).

9	Cervix Uteri Cancer (C53)	Corpus Uteri Cancer (C54)	Ovary Cancer (C56)
2000	167.7	36.6	143.8
2001	149.3	35.7	145.4
2002	152.5	33.8	144.1
2003	146.9	35.6	146.1
2004	146.2	32.1	145.3
2005	139.8	30.8	140.9
2006	136.8	32.4	143.2
2007	138.5	31.5	144.5
2008	125.2	35.4	138.3
2009	125.2	35.0	137.8
2010	121.4	35.7	139.0
2011	113.6	36.2	135.8
2012	113.1	38.9	124.2
2013	105.0	36.2	132.7
2014	97.6	36.5	134.9
2015	92.3	40.9	131.0
2016	87.9	40.8	117.4
2017	92.4	41.8	115.4
2018	86.0	40.6	121.0
2019	82.7	41.0	117.8
2020	77.8	39.4	114.3
2021	75.0	35.4	109.5
AAPC	−3.7 *	0.9 *	−1.3 *
(95% CI)	(−3.9; −3.4)	(0.5; 1.4)	(−1.6; −1.1)

* *p* < 0.05.

## Data Availability

The data presented in this study are available on request from the corresponding author.

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
