# Peer review of "Trends in Mortality Due to Malignant Neoplasms of Female Genital Organs in Poland in the Period 2000–2021—A Population-Based Study"

_cancers, 2024, doi:10.3390/cancers16051038_

Round 1
Reviewer 1 Report (Previous Reviewer 1)
Comments and Suggestions for Authors
The authors have completed an extensive revision of their manuscript and addressed all of my concerns previously raised in my review. I have no additional concerns that require addressing.
I did want to comment on my suggestion about assessing differences in the rates by education or geographic location. This could be assessed using rate ratios and their corresponding confidence intervals but not statistical tests based on samples. I am not requesting this to be done but could be done at the discretion of the authors.
Comments on the Quality of English LanguageGenerally the paper is well written and easy to read. There were some minor English corrections that I saw that could be corrected before publication.
Author Response
We are very grateful for the thorough analysis of our work. We are also confident that any corrections made based on the Reviewer's comments have contributed to improving the quality of our manuscript.
Reviewer 2 Report (Previous Reviewer 2)
Comments and Suggestions for Authors
Most of the weaknesses of the first version of the paper have been resolved.
Analyses of the entire MNFGO mortality rates continue to be not very significant. It is not very correct, since cancers in this group are very heterogeneous for risk factors, etiology, approach, age of death and mortality trends (figure 2a does not appear useful).
The same for statistics for “other part of the uterus”: you have clarified that C55 (unspecified uterus) are few. It looks better to show cervix and corpus uteri mortality separately, since C55 cases (which could be “body”, but also “cervix”) are irrelevant “missing information”.
Showing (tab 1) and commenting joinpoints when they are not statistically significant is not important and they generate only impressions, but not sound hypotheses.
The intention to increase cervical screening coverage from 12% to 80% in a few years is extremely ambitious. It could be very interesting to know some more about the program for hereditary cancer screening and control.
As already suggested, an analysis of years of life lost (YLL) could have added something new and interesting deepening about the real age-mortality burden in Poland, respect to the observed and already published data (https://onkologia.org.pl/en/report). Polish Cancer Registry shows great differences in MNFGO mortality within the Country which could have been interesting to deepen.
In row 37-38 a misleading statement remains: MNFGO mortality does not “often” affect young women. In the class 35-39 yrs it starts to increase, but it “often” affects older women (with a peak over the age of 70).
Author Response
We are very grateful for the thorough analysis of our work. We are also confident that any corrections made based on the Reviewer's comments have contributed to improving the quality of our manuscript.
Manuscript has been redrafted in accordance with the Reviewer's suggestions. Below, we refer in detail to the individual comments.
Analyses of the entire MNFGO mortality rates continue to be not very significant. It is not very correct, since cancers in this group are very heterogeneous for risk factors, etiology, approach, age of death and mortality trends (figure 2a does not appear useful).
As suggested by the Reviewer, Figure 2a and the paragraph of the Results section referring to this graph have been deleted.
The same for statistics for “other part of the uterus”: you have clarified that C55 (unspecified uterus) are few. It looks better to show cervix and corpus uteri mortality separately, since C55 cases (which could be “body”, but also “cervix”) are irrelevant “missing information”.
Fragments describing trends in mortality due to C55 were removed from the text and the wording of Limitations has been slightly changed.
Showing (tab 1) and commenting joinpoints when they are not statistically significant is not important and they generate only impressions, but not sound hypotheses.
We agree with the Reviewer that commenting on statistically insignificant changes in trends in the text is unnecessary and therefore we have removed several sentences.
However, we disagree that non-significant APC and AAPC values should be removed from Table 1. We have clearly marked with an asterisk which values are and are not statistically significant, and 95% confidence intervals are also added.
The intention to increase cervical screening coverage from 12% to 80% in a few years is extremely ambitious. It could be very interesting to know some more about the program for hereditary cancer screening and control.
The methods outlined in the National Oncology Strategy to achieve the 80% target have been added, as well as a fragment about the program for hereditary cancer screening and control.
As already suggested, an analysis of years of life lost (YLL) could have added something new and interesting deepening about the real age-mortality burden in Poland, respect to the observed and already published data (https://onkologia.org.pl/en/report). Polish Cancer Registry shows great differences in MNFGO mortality within the Country which could have been interesting to deepen.
According to the Reviewer's suggestion, Potential Years of Life Lost (PYLL) were calculated due to the three most important causes of death in the MNFGO group for the years 2000-2021 (Table 2). An appropriate fragment has been added in the Abstract, Material and Methods, Results and Discussion sections.
Analyzing the differences for 16 Polish voivodeships and trying to consider the reasons for this difference would probably be interesting. However, we believe that this is such an extensive topic that it should be described in a separate manuscript, addressed especially to Polish readers.
In row 37-38 a misleading statement remains: MNFGO mortality does not “often” affect young women. In the class 35-39 yrs it starts to increase, but it “often” affects older women (with a peak over the age of 70).
The wording of the sentence has been changed to the following:
"MNFGOs are not only an important health problem but also a social problem because mortality due to these causes begins to increase from the age of 35 and often concerns women, professionally active and raising children."
This manuscript is a resubmission of an earlier submission. The following is a list of the peer review reports and author responses from that submission.
Round 1
Reviewer 1 Report
Comments and Suggestions for Authors
Please see attached document.

Quality of English language is generally fine. I noted a few minor edits.
Reviewer 2 Report
Comments and Suggestions for Authors
Main observations
General/methods
The title of the study is not coherent with its topic since it does not investigate “prematurity” in itself (loss of years of life) but mortality rates and their time-trends for cancer of female genital organs.
The study presents the mortality data and their time-trends for women’s genital tract in Poland (2000-2021). Except for 2021, the same data and trends (1999-2020) have already been published by the Polish Cancer Registry and they are available at https://onkologia.org.pl/en/report: This paper adds only some general comments to that public data, but no new information.
The “results” section includes some data on the effect of urbanization and education in determining female tract cancer mortality. However, in the “material and methods” section, none is said about the sample, covariates and methods used to investigate this topic.
The MNFGO group includes cancers with different age and birth-cohorts at risk, risk factors, etiology, clinical approach, age of death (and mortality time-trends). Analyses concerning the whole MNFGO group do not appear significant. It would have been interesting to analyze the years of life lost and the specific role played on mortality by incidence, late diagnosis and access to therapies, stratified by single disease and socio-economic status.
Discussion
No interpretation was given to the opposite trends in ovarian cancer mortality, as well as different trends in corpus uteri.
None is said about the possible repercussions of decrease over time of unspecified cancers of the uterus (likely due to an improvement in recording accuracy). Could it bias the presented cervical and corpus uteri time-trends?
Oversights:
Row 38: The sentence: “MNFGO affect young women (35-59 anni) is not correct”: the incidence grows significantly since the 35-59 yrs group, but (in western Countries) each anatomic site examined has its peak of incidence in women older than 70 yrs.
Row 180: vaccination can allow to prevent cervical cancer, but not its “early diagnosis”, nor its “eradication” (vaccines don’t cover all high-risk types of HPV)
Rows 113-14: the total number of deaths for corpus uteri and ovary of supplementary table 1 don’t correspond to those indicated in these rows and in the abstract.
Row 131: During the period mortality “halved” (not “doubled”)
Rows 232-3: to which country/countries do this data refer? Please indicate a bibliographical reference.
Comments on the Quality of English LanguageMinor editing of English language required